# 1 History of anthropogenic Phosphorus inputs (HaPi) to the terrestrial

## 2 biosphere from 1860 to 2020

- Zihao Bian<sup>1,2</sup>\*, Hao Shi<sup>3</sup>\*, Rui Li<sup>4</sup>, Fei Lun<sup>5</sup>, Francesco N. Tubiello<sup>6</sup>, Nathaniel D.
- Mueller<sup>7,8</sup>, Shiyu You<sup>3</sup>, Rong Hao<sup>1,2</sup>, Jiageng Ma<sup>3</sup>, Longhui Li<sup>1,2</sup>, Changchun Huang<sup>1,2</sup>,
- Bing He<sup>9</sup>, Yuanzhi Yao<sup>4</sup>, Hanqin Tian<sup>10,11</sup>\*

- <sup>1</sup>State Key Laboratory of Climate System Prediction and Risk Management, Nanjing
- Normal University, Nanjing 210023, China.
- <sup>2</sup>School of Geography, Nanjing Normal University, Nanjing 210023, China
- <sup>3</sup>State Key Laboratory for Ecological Security of Regions and Cities, Research Center
- for Eco-Environmental Sciences, Chinese Academy of Sciences, Beijing 100085,
- 13 China
- <sup>4</sup>Key Laboratory of Geographic Information Science of the Ministry of Education,
- School of Geographic Sciences, East China Normal University, Shanghai 200241,
- China
- <sup>5</sup> College of Land Science and Technology, China Agricultural University, Beijing
- 100193, China
- <sup>6</sup>Statistics Division, Food and Agriculture Organization of the United Nations, Via
- Terme di Caracalla, Rome 00153, Italy
- <sup>7</sup>Department of Ecosystem Science and Sustainability, Colorado State University,
- Fort Collins, CO 80523, USA
- \*Department of Soil and Crop Sciences, Colorado State University, Fort Collins, CO
- 80523, USA
- <sup>9</sup>School of Computer Science, Chengdu University of Information Technology,
- Chengdu 610225, China
- <sup>10</sup>Center for Earth System Science and Global Sustainability, Schiller Institute for
- Integrated Science and Society, Boston College, Chestnut Hill, MA 02467, USA
- <sup>11</sup>Department of Earth and Environmental Sciences, Boston College, Chestnut Hill,
- MA 02467, USA

\*Corresponding authors:

Zihao Bian (zhbian@njnu.edu.cn)

Hao Shi (haoshi@rcees.ac.cn)

Hanqin Tian (hanqin.tian@bc.edu)

#### Abstract

33

34 Nitrogen and Phosphorus (P) are essential nutrients for sustaining life on Earth and have been increasingly applied in global agriculture to meet the growing demand for food 35 36 production. Quantifying the spatial and temporal dynamics of nutrient inputs to the terrestrial biosphere is crucial for analyzing nutrient flows in crop-livestock systems, 37 38 managing nutrient resources sustainably, and mitigating nutrient-related environmental impacts. Here, built upon our previous work mapping global nitrogen inputs (History 39 of anthropogenic Nitrogen inputs, HaNi), this study presents the History of 40 anthropogenic Pinputs (HaPi) dataset, a comprehensive quantification of human-driven 41 42 P fluxes to terrestrial ecosystems. HaPi covers the period from 1860 to 2020 and has a spatial resolution of 5 arc-minutes (about 10 km at the equator) with an annual time-43 step. This harmonized dataset integrates seven components, including P fertilizer 44 application on croplands and pastures, manure P application on croplands and pastures, 45 manure P deposition on pastures and rangelands, and atmospheric P deposition. The 46 results reveal that the global total P input increased from 3.8 Tg yr<sup>-1</sup> in the 1860s to 40.9 47 Tg yr<sup>-1</sup> in the 2010s, with mineral fertilizer and livestock manure contributing equally 48 to the increase. Regional patterns have shifted significantly over the study period, with 49 China, South Asia, and Brazil surpassing Europe and the USA as the regions with the 50 highest P inputs in recent decades. Furthermore, mineral fertilizers dominate P inputs 51 in most industrialized countries in the Northern Hemisphere, whereas manure P remains 52 the primary source in many countries of the Southern Hemisphere. The HaPi dataset 53 improves P mass budget calculations and provides essential forcing data for empirical 54 55 or mechanistic models, supporting critical research in agricultural nutrient management, 56 water quality control, and assessments of the coupled human-Earth system. The dataset is available at https://doi.org/10.6084/m9.figshare.29930279.v1 (Bian et al., 2025). 57

58

#### 1 Introduction

60

61 The global nitrogen and phosphorus (P) cycle has been unprecedently disturbed in the Anthropocene, posing significant risks to planetary boundaries and societal 62 63 sustainability (Steffen et al., 2015). Driven by rising food demand, mining of phosphate rocks for mineral P fertilizers and livestock feed has significantly increased (Cordell et 64 65 al., 2009). Anthropogenic activities have substantially intensified global P flows, contributing to half of global agricultural soil P uptake in recent decades (Demay et al., 66 2023) and tripling the P mobilization in the land-water continuum (Yuan et al., 2018). 67 However, agriculture's growing demand for mineral P fertilizers raises critical concerns 68 69 regarding the depletion of non-renewable phosphate rock reserves. Moreover, the unevenly distributed phosphate rock resources and potential geopolitical conflicts 70 threaten the resilience of agricultural systems (Barbieri et al., 2022; Elser and Bennett, 71 72 2011). To mitigate these challenges, a better understanding of the spatial and temporal distribution of P supply and demand would support improved assessment of future P 73 74 requirements for crop and livestock production (Sattari et al., 2012). 75 The historical and current management of anthropogenic P in croplands and livestock production has resulted in serious environmental issues. In particular, the global P flux 76 77 from terrestrial to aquatic ecosystems has been amplified by inefficient agricultural applications, which accelerated P losses to rivers, lakes, and oceans via water transport. 78 Generally, P is the major limiting nutrient for phytoplankton growth in freshwaters, 79 since its easy adsorption to particles reduces its biological availability (Conley et al., 80 2009). However, worldwide mineral P fertilizer applications and related runoff have 81 resulted in elevated levels of bioavailable P in freshwaters, causing widespread 82 eutrophication and ecological damage to freshwater systems (Carpenter, 2005; Fink et 83 al., 2018). Beyond contemporary P inputs, legacy P accumulated in soils from historical 84 fertilizer and manure applications continues to leach into aquatic environments. This 85 reduces the efficacy of long-term conservation efforts aimed at improving water quality 86 (Stackpoole et al., 2019). Spatial-temporal patterns of agricultural legacy P and 87

88 associated water pollution are fundamentally driven by historical fertilizer and manure applications. Consequently, understanding the quantity, sources, and trends of P 89 pollution is useful information needed for advancing effective water quality 90 management strategies. 91 To address the challenges of food security and environmental issues related to 92 93 anthropogenic P use, accurately quantifying global anthropogenic P inputs since the pre-industrial era is critical for evaluating historical trajectories of soil P fertility, P use 94 efficiency (PUE), and P pollution. Many efforts have been carried out to develop global 95 P input datasets (Lu and Tian, 2017; Lun et al., 2018; Ringeval et al., 2024). The Food 96 97 and Agriculture Organization (FAO) and the International Fertilizer Association (IFA) provide widely adopted country-level mineral P fertilizer data in agricultural land (FAO, 98 2024). This data has been used to evaluate P budgets and surpluses at the country level. 99 For example, Zou et al. (2022) used FAO P fertilizer data to evaluate national P budget 100 and PUE. However, FAO's manure data only covers national nitrogen excretion, 101 102 necessitating estimation of manure P via N:P ratios. For spatially explicit modeling, several gridded datasets have emerged. Mueller et al. (2012) generated crop-specific P 103 fertilizer maps at a global scale circa 2000. Bouwman et al. (2013) developed gridded 104 P fertilizer and livestock manure data by combining FAO fertilizer use, animal 105 production data, and multiple ancillary datasets. This P input dataset has been applied 106 in the IMAGE-Global Nutrient Model to assess global soil P pools, crop uptake, and 107 riverine P fluxes (Beusen et al., 2016). More recently, gridded data of P application rates 108 for 173 crops, at 5 km resolution and the year 2020, have been published (Nguyen et 109 110 al., 2024). Furthermore, Lu and Tian (2017) developed an annual global P fertilizer dataset from 1961 to 2013 at a resolution of 0.5°×0.5°, based on the IFA national P 111 fertilizer statistics. Bouwman's dataset and Lu's P fertilizer data have been used to 112 develop the global dataset on P in agricultural soil, GPASOIL-v0 (Ringeval et al., 2017) 113 and GPASOIL-v1 (Ringeval et al., 2024), respectively. GPASOIL-v1 also provides 114 manure P application rates on cropland and pasture based on a global constant N:P ratio 115 and spatially explicit manure N data developed by Xu et al. (2018) and Zhang et al. 116

(2017). Nevertheless, inconsistencies exist in temporal coverage, spatial resolution, spatial allocation algorithms, and the baseline land use data among different P input 118 datasets, which may propagate uncertainties in assessments of soil P fertility, legacy P 119 120 accumulation and its environmental impacts. To resolve these issues, we propose to reconstruct a harmonized History of Anthropogenic Phosphorus Inputs (HaPi) dataset, 121 which will integrate available FAO statistics, historical land use, grid-level manure N:P 122 123 ratios, and atmospheric P deposition within a consistent spatiotemporal framework. In our recent study, we have developed the History of anthropogenic Nitrogen inputs 124 (HaNi) dataset, which consists of N fertilizer, manure N, and atmospheric N inputs on 125 126 cropland, pasture, and rangeland (i.e., grassland) from 1860 to 2020 (Tian et al., 2022). Since its publication, HaNi has been widely adopted to estimate global N<sub>2</sub>O emission 127 (Tian et al., 2024), NH<sub>3</sub> concentrations (Ma et al., 2025) and emissions, and N loading 128 (Dai et al., 2023). Complementary to the HaNi data, the HaPi dataset provides P 129 fertilizer/manure application to cropland, P fertilizer/manure application to pasture, 130 131 manure P deposition on pasture/rangeland, and atmospheric P deposition. The HaPi dataset advances over previous datasets in five aspects: (1) the impact of dynamic crop 132 rotation is considered when allocating P fertilizer on cropland; (2) the annual spatial-133 explicit manure N:P ratio data is developed to generate manure P data based on manure 134 N inputs; (3) all P inputs are at an annual step from 1860 to 2020 and a high spatial 135 resolution of 5 arc-min; (4) beyond agricultural land, P inputs to all terrestrial 136 ecosystems are estimated; and (5) consistent baseline land use maps are used when 137 allocating fertilizer and manure P to cropland, pasture, and rangeland. The HaPi dataset 138 139 is anticipated to improve P mass budget calculation and serve as forcing data for 140 empirical or mechanistic models, thereby supporting diverse research on soil fertility, 141 water pollution, P resource sustainability, and food security, etc.

142

#### 2 Methods

#### 2.1 Fertilizer application on cropland and pasture

The fertilizer P application maps are developed by allocating country-level inventory data into grid cells according to land use maps, crop rotation maps, and crop-specific fertilizer application data (Table 1). Annual national P fertilizer inputs on agricultural land during 1961-2020 were obtained from FAOSTAT (2024). We first separated total P fertilizer use into P application on cropland and pasture by assuming fractions of P used for cropland and pasture are the same as those of N fertilizer, following the method used in Zou et al. (2022). The fraction data were adopted from Einarsson et al. (2021) and Lassaletta et al. (2014). Then, the country-level P fertilizer data were spatially distributed with cropland and pasture maps following the workflow in Figure 1.

To determine the P fertilizer application on cropland, we considered 18 dominant crop

To determine the P fertilizer application on cropland, we considered 18 dominant crop types (wheat, maize, rice, barley, millet, sorghum, soybean, sunflower, potato, cassava, sugarcane, sugar beet, oil palm, rapeseed, groundnut, cotton, and rye) and crop-specific harvested areas to generate crop-area-weighted average of P fertilizer application in each grid during 1961-2020, which is calculated as follows:

$$\overline{F_{crop,y,g}} = \frac{\sum_{i} (F_{crop,i,2000} \times AH_{i,y})}{\sum_{i} AH_{i,y}}$$
 (1)

where  $\overline{F_{crop,y,g}}$  is the crop-area-weighted average of P fertilizer application rate in a cropland grid g (g P m<sup>-2</sup> cropland yr<sup>-1</sup>) in year y,  $F_{crop,i,y}$  and  $AH_{i,y}$  are P fertilizer application rate (g P m<sup>-2</sup>) and harvested area (g matrix g matrix, respectively, for crop type g in year g matrix, g matrix g matrix

- (temperature and precipitation) (Fick and Hijmans, 2017), soil information (FAO and
- IIASA, 2023), and topography. In each year, crop-specific distribution from SPAM2010
- (Yu et al., 2020) was used to start an interactive training process to obtain a stable
- machine learning model that link gridded crop fractions to covariates, with a constraint
- that all crop areas in a grid cell in a growing season could not exceed the total available
- land area of that grid.
- Then we calculated the potential P fertilizer application on cropland in a specific
- country and used it to divide the corresponding FAO statistics:

$$R_{crop,y,j} = \frac{FAO_{crop,y,j}}{\sum_{g=1}^{g=n \text{ in country } j} (\overline{F_{crop,y,g}} \times AC_{y,j,g})}$$
 (2)

$$Pfer_{crop,y,j,g} = \overline{F_{crop,y,g}} \times R_{crop,y,j}$$
 (3)

- where  $R_{crop,v,j}$  represents the regulation ratio (unitless) for the year y and country j,
- $FAO_{crop,y,j}$  is the national total P fertilizer usage (g P yr<sup>-1</sup>) on cropland,  $AC_{y,i,q}$  is the
- cropland area  $(m^2)$  for the year y and grid g of the country. The historical land use data
- (cropland, pasture, rangeland) covers 1860-2020 and has an annual time-step and a 5
- arcmin spatial resolution, developed by reconciling the HYDE 3.2 and LUHv2 datasets
- (Hurtt et al., 2020; Klein Goldewijk et al., 2017; Tian et al., 2022). The P fertilizer in
- cropland grid g of the country j in year y,  $Pfer_{crop}$  (g P m<sup>-2</sup> yr<sup>-1</sup>), was then calculated
- as the product of  $F_{crop,y,g}$  and  $R_{crop,y,j}$ .
- The earliest year of P fertilizer statistics that the FAO database provides is 1961. Prior
- to the widespread adoption of phosphate rock as a fertilizer in the 1940s, guano—
- accumulated bird droppings over millennia—and human excreta were utilized as
- fertilizers for food crops. The historical P fertilizer data during 1860-1960 were
- generated based on data reported by Cordell et al. (2009), which provide global total P
- fertilizer usage estimates back to 1800. Annual relative change rates of global P
- fertilizer were first calculated based on the estimates from Cordell et al. (2009). These
- change rates were then applied to the spatially explicit P fertilizer distribution in the

reference year of 1961 to reconstruct annual global P fertilizer maps for the period 1860-1960. This approach assumes that temporal dynamics of P fertilizer data before 1961 were primarily driven by global-scale trends.

Figure 1. Workflow of developing data for P fertilizer application on cropland. The blue box represents the annual data during 1961-2020, and the orange box represents 201

the static variable.

204

Table 1. Main data sources utilized in HaPi development

|            |             |                                   | •                     |
|------------|-------------|-----------------------------------|-----------------------|
| Data       | Data Source | Dataset                           | Reference             |
| Products   |             |                                   |                       |
| Fertilizer | FAOSTAT     | Annual country-level P fertilizer | FAOSTAT (2024)        |
|            |             | to land from 1961 to 2020;        |                       |
|            |             | Crop-specific harvested area      |                       |
|            | EARTHSTAT   | Fertilizer rates for major crops  | Mueller et al. (2012) |
|            | SPAM2010    | Crop distribution maps            | Yu et al. (2020)      |
|            |             |                                   |                       |

|             | Hyde3.2/LUHv2     | Cropland, Pasture, and rangeland | Hurtt et al. (2020);  |  |
|-------------|-------------------|----------------------------------|-----------------------|--|
|             |                   | area from 1860 to 2019           | Klein Goldewijk et    |  |
|             |                   |                                  | al. (2017)            |  |
|             | Cordell et al.    | Global total P fertilizer usage  | Cordell et al. (2009) |  |
|             | (2009)            | from 1860 to 1960                |                       |  |
| Manure      | HaNi              | Manure Nitrogen inputs from      | Tian et al. (2022)    |  |
|             |                   | 1860 to 2020                     |                       |  |
|             | GLW3              | Livestock distribution maps      | Gilbert et al. (2018) |  |
|             | FAOSTAT           | Annual country-level livestock   | FAOSTAT (2024)        |  |
|             |                   | statistics from 1961 to 2020     |                       |  |
|             | Lun et al. (2018) | Animal-specific manure P:N       | Lun et al. (2018)     |  |
|             |                   | ratios                           |                       |  |
| Atmospheric | НЕМСО             | Natural P emissions (mineral     | Meng et al. (2021);   |  |
| deposition  |                   | dust, sea-salt aerosols, and     | Weng et al. (2020);   |  |
|             |                   | volcanic eruptions)              | Carn (2019)           |  |
|             | EDGAR             | Anthropogenic P emissions        | Crippa et al. (2023)  |  |
|             | GFED v4           | P emissions from wildfire        | Randerson et al.      |  |
|             |                   |                                  | (2015)                |  |
|             | ERA5              | Climate data (temperature,       | Muñoz-Sabater et al.  |  |
|             |                   | precipitation, wind speed, etc.) | (2021)                |  |
|             | MERRA2            | Aerosol Optical Thickness        | Randles et al. (2017) |  |
|             | GIMMS LAI4g       | Leaf area index                  | Cao et al. (2023)     |  |
|             |                   |                                  |                       |  |

### 2.2 Manure application on cropland/pasture and deposition on pasture/rangeland

The grid-level manure P inputs, including manure application on cropland, manure application on pasture, manure deposition on pasture, and manure deposition on rangeland, were generated by multiplying the corresponding manure N inputs from

228229

230

HaNi (Tian et al., 2022) and manure P:N ratios (Figure 2):

$$Pman_{\nu,q} = Nman_{\nu,q} \times RPN_{\nu,q} \tag{4}$$

- where  $Pman_{y,g}$  and  $Nman_{y,g}$  indicate the P and N manure application or deposition on grid g in year y, respectively. The HaNi dataset offers global manure input data on cropland, pasture, and rangeland during 1860-2020 at a resolution of 5 arcmin.  $RPN_{y,g}$
- represents the manure P:N ratio in grid g.

$$RPN_{y,g} = \frac{\sum_{t} APN_{y,t} \times Anum_{y,g,t}}{\sum_{t} Anum_{y,g,t}}$$
 (5)

$$Anum_{y,g,t} = GLMnum_{g,t} \times \frac{LFAO_{y,t,j}}{\sum_{g=1}^{g=n \text{ in country } j} GLMnum_{g,t}}$$
 (6)

The annual grid-level manure P:N ratios  $RPN_{y,g}$  were estimated as the animal-number weighted sum of these animal-specific P:N ratios within grids. We obtained manure P:N ratios  $APN_{y,t}$  for each animal species t from Lun et al. (2018). Livestock distribution data  $Anum_{y,g,t}$  were calculated using national FAO livestock statistics and the Global Livestock of the World 3 (GLW3) database (Gilbert et al., 2018). GLW3 provides global livestock populations  $GLMnum_{g,t}$  at a 5-arcmin resolution for cattle, buffaloes, horses, sheep, goats, pigs, chickens, and ducks, but the data are available for a single reference year. Therefore, the annual country-level livestock population data  $LFAO_{y,t,j}$  from FAOSTAT were employed to extend the GLW3 dataset into a time series of livestock distribution maps spanning from 1961 to 2020 following the method used in equations 2-3. Based on this approach, a spatially explicit manure P:N ratio dataset covering the period 1961–2020 was constructed and subsequently used to estimate manure P inputs from corresponding N inputs.

**Figure 2.** Workflow of developing manure P input data. The blue box represents the annual data during 1961-2020, and the orange box represents the static variable.

### 2.3 Atmospheric P deposition

Atmospheric P deposition originates from both natural and anthropogenic sources and was estimated by combining a P emission inventory, an atmospheric chemical model, and an algorithm (Figure 3 and Table 1). Natural P emissions primarily comprise dust, sea salt, Primary Biological Aerosol Particles (PBAPs), volcanoes, and wildfires. In this study, emissions from dust, sea salt, and volcanoes were derived from the GEOS-Chem HEMCO data (Meng et al., 2021; Weng et al., 2020; Carn, 2019; Carn et al., 2015; Ge et al., 2016). PBAPs emissions were estimated by utilizing specific humidity data and LAI data according to the method used in Myriokefalitakis et al. (2016). Wildfire emission data were sourced from the GFED4 dataset (Randerson et al., 2015). Meanwhile, anthropogenic emissions were estimated using monthly sector-specific data from the EDGAR database (Crippa et al., 2023). By integrating all the emission sources, we developed a global atmospheric P emission inventory, which was subsequently used as input for the GEOS-Chem model (Feng et al., 2021; Li et al., 2024) to simulate global atmospheric P deposition from 2001 to 2019. GEOS-Chem is a global

three-dimensional atmospheric chemical model driven by meteorological inputs provided by the Goddard Earth Observing System (GEOS) of the NASA Global 251 Modeling and Assimilation Office. It has been widely applied to address various 252 253 atmospheric composition issues. The modeled P deposition by GEOS-Chem was at a 254 resolution of 2.5°×2°. 255 To obtain the atmospheric P deposition at a high resolution, the dissever algorithm (Malone et al., 2012; Roudier et al., 2017), which is a mass-conserving spatial 256 257 downscaling method, was applied to the GEOS-Chem outputs. First, the coarse resolution P deposition data was resampled to 0.1 ° × 0.1 °; Second, the dissever 258 algorithm was used to iteratively train an XGBoost model for eight different regions, 259 including Africa, Asia excluding China and India, China, Europe, India, North America, 260 Oceania, and South America, to retrieve resampled P deposition and fine-resolution 261 environmental variables. These environmental variables included ERA5 climate factors 262 (wind components, temperature, surface net solar radiation, surface pressure, and 263 264 monthly total precipitation) (Muñoz-Sabater et al., 2021), LAI (Cao et al., 2023), 265 Aerosol Optical Thickness (AOT, Randles et al., 2017), and P emission data. The 266 iterative model training process was terminated when the improvement in performance 267 between successive generations fell below a predefined threshold. Model performance was assessed using the Mean Absolute Error (MAE; unit: g ha<sup>-1</sup>), calculated for each 268 269 model's target region across all terrestrial grid cells using the test dataset. The 270 convergence threshold for MAE was initially set to 0.001 g ha<sup>-1</sup>. Using this approach, we generated global atmospheric P deposition estimates at a spatial resolution of 0.1° 271 for the period 2000-2019. To ensure consistency with the HaPi dataset, the 0.1° data 272 were subsequently resampled to a 5 arc-min resolution and temporally extended back 273 274 to 1900 using annual P deposition change rates derived from Ringeval et al. (2024).

**Figure 3.** Workflow of developing atmospheric P input data. Where PBAPs represent Primary Biological Aerosol Particles. Prec refers to precipitation. T2m indicates the 2-meter air temperature, while T2m max and T2m min refer to the maximum and minimum air temperatures at 2-meter height. U10 and V10 denote the eastward and northward wind speeds at 10-meter height, respectively. AOT stands for Aerosol Optical Thickness. SP represents Surface Pressure, and SSRD refers to Surface Solar

### 3 Results

#### 3.1 Global total P inputs

Radiation Downwards.

Global anthropogenic P inputs have undergone dramatic changes since the Industrial Revolution, increasing from 3.8 Tg P yr<sup>-1</sup> in the 1860s to 40.9 Tg P yr<sup>-1</sup> in the 2010s, a nearly 11-fold increase (Figure 4). The most rapid acceleration occurred between 1945 and 1989, coinciding with post-war agricultural intensification. The amplified usage of mineral fertilizer and livestock manure P contributed nearly equally, over the study period, i.e., 52% and 46% of the increase in TP inputs to the terrestrial biosphere, respectively. The composition of P inputs shifted significantly over this period (Table

2). Prior to the 1980s, livestock manure accounted for over half of total P (TP) inputs. However, by the 2010s, mineral fertilizer P in TP inputs surpassed manure P for the first 293 time. Atmospheric P deposition, while relatively stable, declined from 50% to only 6% 294 of TP inputs from the 1860s to the 2010s, due to the marked increase in anthropogenic 295 296 inputs. The intensified application of P fertilizer on cropland was the primary driver behind the increased TP inputs after 1945. 297 Regionally, the TP inputs initially increased in Europe and eastern USA between the 298 1860s and the 1910s, primarily due to enhanced manure inputs (Figures 5-6). 299 Subsequently, P input hotspots emerged in Europe, driven by the rising use of P fertilizer. 300 Europe (5.1 Tg P yr<sup>-1</sup>) and the USA (2.8 Tg P yr<sup>-1</sup>) were the top two regions with the 301 highest P inputs in the 1960s. After the 1980s, TP inputs in Europe declined quickly, 302 while leveling off in the USA. By the 2010s, new hotspots emerged in eastern China, 303 northern India, southern Brazil, and eastern Africa, sourcing from the widespread 304 application of mineral P fertilizer and the expansion of livestock production in these 305 regions. Notably, TP inputs in China began to decrease around 2010, whereas inputs in 306 South Asia and Brazil maintained growth trends. Livestock manure remains the 307 dominant P source in most regions of the South Hemisphere (Africa, South America, 308 Oceania), while P fertilizer plays a more important role in industrialized regions of the 309 North Hemisphere (Asia, Europe, North America). In addition to Europe, TP inputs in 310 Russia, Korea, and Japan (KAJ), and Central Asia (CAS) also declined substantially, 311 largely due to the reduction in P fertilizer usage. Manure P inputs continued to increase 312 in Africa, South/Southeast Asia, and Central America (CAM), but have shifted to a 313 314 decreasing trend in Oceania, Europe, and Russia in recent decades.

318

Figure 4. Annual changes of anthropogenic phosphorus inputs to terrestrial ecosystems from 1860 to 2020.

**Table 2.** Decadal mean P inputs to the terrestrial biosphere (Tg P yr<sup>-1</sup>)

| Decade | $P_{fer}$ | Pfer    | P <sub>man</sub> | P <sub>man</sub> | P <sub>man</sub> | P <sub>man</sub> | P <sub>dep</sub> | Total |
|--------|-----------|---------|------------------|------------------|------------------|------------------|------------------|-------|
|        | Crop      | Pasture | Crop             | App              | Dep              | Dep              |                  |       |
|        |           |         |                  | Pasture          | Pasture          | Range            |                  |       |
| 1860s  | 0.1       | 0.0     | 0.5              | 0.2              | 0.7              | 0.3              | 1.9              | 3.8   |
| 1910s  | 0.6       | 0.0     | 1.3              | 0.4              | 1.8              | 1.1              | 1.9              | 7.1   |
| 1960s  | 6.2       | 0.4     | 2.8              | 0.7              | 5.1              | 3.5              | 2.0              | 20.6  |
| 1970s  | 10.5      | 0.9     | 3.3              | 0.8              | 5.9              | 3.9              | 2.2              | 27.5  |
| 1980s  | 13.9      | 1.1     | 3.7              | 0.9              | 6.7              | 4.2              | 2.3              | 32.8  |
| 1990s  | 13.2      | 1.0     | 3.9              | 0.8              | 7.3              | 4.4              | 2.3              | 33.1  |
| 2000s  | 14.7      | 1.3     | 4.0              | 0.8              | 7.9              | 4.8              | 2.4              | 35.8  |
| 2010s  | 17.9      | 1.5     | 4.3              | 0.8              | 8.8              | 5.0              | 2.6              | 40.9  |

Note: The following abbreviations are used in the table:  $P_{fer} - P$  fertilizer,  $P_{man}$  – manure P,  $P_{dep}$ 

- Atmospheric P deposition, App - Application, and Dep - Deposition.

323324

Atmospheric P deposition.

**Figure 5.** Spatial patterns in total Phosphorus input in the (a) 1860s, (b) 1910s, (c) 1960s, and (d) 2010s. The labels in the inset pet charts represent the percentage of each component: Fc – P fertilizer applied to cropland, Fp – P fertilizer applied to pasture, Mc – Manure P application on cropland, Map – Manure P application on pasture, Mdp – Manure P deposition on pasture, Mr – Manure P deposition on rangeland, Ad –

**Figure 6.** Annual variations of P inputs in 18 regions during 1860-2020. The 18 regions are the Canada (CAN), USA (USA), Europe (EU), Central Asia (CAS), Russia (RUS), Korea and Japan (KAJ), China (CHN), South Asia (SAS), Southeast Asia (SEAS),

Oceania (OCE), Middle East (MIDE), Southern Africa (SAF), Equatorial Africa

(EQAF), Northern Africa (NAF), southwestern South America (SSA), Brazil (BRA),

northern South America (NSA), and Central America (CAM).

### 3.2 P fertilizer inputs on cropland and pasture

The global total P fertilizer inputs surged from 6.6 Tg P yr<sup>-1</sup> in the 1960s to 19.4 Tg P yr<sup>-1</sup> in the 2010s, with cropland receiving over 90% of these inputs (Figure 7). Annual P fertilizer applied to croplands increased rapidly at a rate of 0.4 Tg yr<sup>-2</sup> during 1961-1989, then declined sharply until 1995 before resuming an upward trajectory. Despite this global growth, significant regional disparities emerged (Figures 6 and 8). Europe and the USA were the dominant regions for early P fertilizer consumption, accounting for 38% and 24% of global usage, respectively, in the 1960s. By the 2010s, China (30%) and South Asia (20%) had become the predominant consumers of P fertilizers, driven by intensive agricultural expansion. Regional shifts in P fertilizer inputs were pronounced. In the USA, P fertilizer application to cropland decreased from 2.0 Tg P yr<sup>-1</sup> in the 1970s to 1.5 Tg P yr<sup>-1</sup> in the 2000s. In Europe, P fertilizer usage peaked in the 1980s but decreased by 66% in the 2010s. The increase in P fertilizer usage in China and South Asia contributed to 44% and 29% of the global total increase in P fertilizer application on cropland from the 1960s to the 2010s, respectively. However, China's usage plateaued in the 2010s, while South Asia's continued to increase rapidly. In the 2020s, hotspot regions with high P fertilizer application rates (>3 g P m<sup>-2</sup> yr<sup>-1</sup>) were mainly distributed in eastern China, northern India, and southern Brazil. The P fertilizer application rates on cropland were consistently low across most areas in Africa over the whole study period, accounting for only 4% of global consumption in the recent decade. Global P fertilizer application on pasture increased from 0.3 to 1.6 Tg P yr<sup>-1</sup> during 1961-2020. Before the 1980s, European countries were the predominant consumers of P fertilizer on pasture (Figure S1). Thereafter, the USA and India significantly increased

their usage, accounting for 23% and 22% of total P fertilizer application on pasture, respectively, in the 2010s. P fertilizer application rate per pasture land was high in India, Japan, and southern Canada, but remained low in most other countries in recent decades.

362363

**Figure 7.** Annual variations and trends in P fertilizer inputs during 1860–2020.

**Figure 8.** Global patterns of P fertilizer application on cropland in the 1960s, 1980s, 2000s, and 2010s.

### 3.3 Manure P inputs on cropland, pasture, and rangeland

Livestock manure served as the primary P source for agricultural soils historically, with global manure P inputs on cropland, pasture, and rangeland increasing from 1.7 to 18.9

Tg P yr<sup>-1</sup> during the period from the 1860s to the 2010s (Figure 9). Pasture received 370 half of total manure P inputs, while cropland and rangeland each shared a similar 371 proportion of the remaining half. Over 1961-2020, manure P deposition on pasture kept 372 increasing at a rate of 0.07 Tg P yr<sup>-2</sup>, while manure P application on pasture was 373 relatively stable. Meanwhile, annual manure inputs on cropland and rangeland both 374 increased, but at relatively slow rates than those on pasture. In the context of rising 375 global manure P inputs, developing countries have demonstrated stronger growth rates 376 377 than developed countries in recent decades. China has exceeded Europe and become the largest consumer of manure P since the 2000s, as manure P inputs in Europe have 378 declined since the 1980s. Over the past four decades, manure usage in North Africa 379 (NAF) and Equatorial Africa (EQAF) has grown at the fastest rate, increasing by 96% 380 and 138%, respectively. In the 2010s, China, South Asia, Brazil, and North Africa, as 381 the top four regions, received 13%, 11%, 11%, and 10% of the global total manure P 382 inputs, respectively. 383 Application rates of manure P per unit area of cropland increased significantly in Asia, 384 Europe, and North and South America since 1860 (Figure 10). Manure P application on 385 386 cropland initially increased in western Europe in the 1910s, with subsequent intensified application occurring in eastern Asia and northern South America by the 2010s. Manure 387 application and deposition rates on pasture were extremely high in South and Southeast 388 Asia over the last century (Figures 11, S2, and S3). The proportion of manure deposition 389 in total manure inputs to pasture gradually increased from 79% to 92% from the 1860s 390 to the 2010s. Prominent manure P deposition on rangeland was observed in South and 391 Southeast Asia, with new hotspots developing in Central Africa, eastern South America, 392 northern China, and Europe in the 2010s (Figure 12). 393

Figure 9. Annual variations and trends in manure P inputs from 1860 to 2020

**Figure 10.** Global patterns of manure P application on cropland in the 1860s, 1910s, 1960s, and 2010s.

**Figure 11.** Global patterns of manure P application and deposition on pasture in the 1860s, 1910s, 1960s, and 2010s.

**Figure 12.** Global patterns of manure P deposition on rangeland in the 1860s, 1910s, 1960s, and 2010s.

# 3.4 Atmospheric P deposition

Atmospheric P deposition is the major P source for natural ecosystems, such as forests and shrubs, but plays a less important role in cropland. Although anthropogenic activities have led to the increase of atmospheric P deposition from 2.0 Tg P yr<sup>-1</sup> in the 1960s to 2.6 Tg P yr<sup>-1</sup> in the 2010s, the proportion of atmospheric P deposition in total P inputs decreased dramatically to only 6% by the 2010s. China was the region with the largest atmospheric P deposition (21% of the global total) in the 2010s, followed by North Africa (15%), the USA (11%), and Europe (8%) (Figure 13). Atmospheric P deposition in northern Africa was primarily derived from natural sources and remained relatively stable during the study period. Influenced by anthropogenic activities, atmospheric P deposition in China has continually increased since the 1960s, while that in Europe has shifted from an increase to a decrease in the 1980s. Atmospheric P deposition was relatively low across most regions, with the exception of several notable hotspots in northern Africa, eastern China, central Europe, and eastern USA.

**Figure 13.** Global patterns of atmospheric P deposition in the 1960s, 1980s, 2000s, and 2010s.

### 4 Discussion

## 4.1 Comparison with previous studies

430

440

442443

448449

The HaPi dataset provides spatially explicit, annual estimates of anthropogenic P inputs to the terrestrial biosphere. This gridded P input dataset reveals subnational heterogeneity, enabling the detection of localized hotspots of high P input that national averages may obscure. When aggregated to the global scale, HaPi aligns well with existing studies, particularly for cropland fertilizer P inputs, which have been extensively evaluated (Table 3). Nearly all these studies rely on the country-level fertilizer inventory data provided by FAO or IFA, which give consistent global totals. However, the HaPi estimates of manure P applied to cropland are lower than previous estimates, whereas manure P inputs in grasslands (pasture and rangeland) are higher than previous estimates. Despite these differences, total manure P inputs in HaPi remain consistent with other studies. The major discrepancy lies in how manure inputs are allocated to cropland and grassland. In previous studies, manure P inputs were calculated by multiplying manure N inputs provided by FAO and P:N ratio in livestock manure products. According to the latest FAOSTAT dataset, around 75% of total manure N inputs are deposited on grassland, and the remaining 25% is applied to cropland (around 21%) and grassland (4%) soils. The majority of manure is directly deposited on grasslands through grazing animal excretion, whereas a smaller proportion is collected and subsequently applied to pastures and croplands as a managed nutrient input. HaPi estimates 23% of manure P was applied on cropland, closely matching FAOSTAT's current methodology. In contrast, Lun et al. (2018) and Sattari et al. (2016) estimated 36% and 38% of total manure P inputs to cropland. Compared to previous studies, the HaPi dataset offers the most comprehensive coverage of anthropogenic P input fluxes, featured by a high spatial resolution and extended historical coverage. These features support the analysis of legacy P accumulation and depletion, and simultaneously provide consistent forcing data for land surface and biogeochemical models. This harmonized P input dataset enhances our ability to assess regional nutrient trends, environmental risks, and management needs at multiple spatial scales.

**Table 3.** Comparison of P input data with other global estimates

| Global P inputs     | Previous studies   | Year      | This study       | Reference             |  |
|---------------------|--------------------|-----------|------------------|-----------------------|--|
|                     | $(Tg\;P\;Yr^{-l})$ |           | $(Tg\;PYr^{-l})$ |                       |  |
| Fertilizer input on | 16.4               | 2002-2010 | 17.6             | Lun et al. (2018)     |  |
| cropland            | 13.8               | 2000      | 13.2             | Bouwman et al.        |  |
|                     |                    |           |                  | (2009)                |  |
|                     | 0.4-15.8           | 1900-2010 | 0.3-17.6         | Zhang et al. (2017b)  |  |
|                     | 4.6-17.5           | 1961-2013 | 4.5-18.0         | Lu and Tian (2017)    |  |
| Fertilizer input on | 0.4                | 2002-2010 | 1.5              | Lun et al. (2018)     |  |
| pasture             |                    |           |                  |                       |  |
| Manure input on     | 1.6-7.2            | 1900-2010 | 1.0-4.3          | Zhang et al. (2017b)  |  |
| cropland            | 7.1                | 2002-2010 | 4.3              | Lun et al. (2018)     |  |
|                     | 6.3                | 2005      | 4.0              | Sattari et al. (2016) |  |
| Manure input on     | 12.7               | 2002-2010 | 14.1             | Lun et al. (2018)     |  |
| grassland (pasture  | 10.2               | 2005      | 13.6             | Sattari et al. (2016) |  |
| and rangeland)      |                    |           |                  |                       |  |
| Total manure        | 17.1               | 2000      | 16.7             | Bouwman et al.        |  |
| input               |                    |           |                  | (2013)                |  |
|                     | 16.5               | 2005      | 17.6             | Sattari et al. (2016) |  |
|                     | 19.8               | 2002-2010 | 18.4             | Lun et al. (2018)     |  |

### 4.2 The implication of changes in manure and fertilizer P inputs

Globally, mineral P fertilizer has surpassed manure P as the largest P input to cropland, while livestock manure remains the dominant P source for pasture and rangeland. Overall, livestock manure is a more significant source of P for terrestrial ecosystems compared to mineral fertilizer. In recent decades, livestock manure contributed over

476

479

485

490

half of the P sources in most regions, especially in Africa where around 90% of total P inputs were derived from manure. As P fertilizer application began to decrease in European countries, manure P application on cropland played an increasingly important role in food production. Grassland, including pasture and rangeland, received over 70% of global manure P inputs and over 40% of total P inputs; therefore, it is critical to take account of grassland when investigating global P balance and P-related environmental issues. For instance, it is estimated that P loading to the Gulf of Mexico originated primarily from manure on pasture and rangeland (37%), followed by corn and soybeans (25%) within the Mississippi River Basin (Alexander et al., 2008). Despite high manure P input, manure excretion is still partly an internal P cycling in the grassland-livestock system since it derives from soil P uptake by grass. Furthermore, global manure P input cannot compensate for the grazing P output in grassland systems because around 23% of manure was transferred from grassland to cropland systems. Due to the limited application of fertilizer to grasslands, global grasslands are facing the challenge of a negative P budget (Sattari et al., 2016). Given the projected increase in livestock production to meet future demand for meat and milk, effectively managing P flows in the whole crop-livestock production systems is critical for the sustainable human P cycle (Bouwman et al., 2013). Socio-economic development clearly drives the long-term changes in total anthropogenic P inputs to agricultural lands (Figure S4). The rapid increase in P fertilizer application on cropland in the latter half of the 20<sup>th</sup> century stimulated crop production but ultimately resulted in a decrease in the P use efficiency. After about 60 years of growth, global total P fertilizer use has leveled off in the 2010s. Europe, previously the largest consumer of P fertilizer, began reducing its usage in the 1980s. China, currently the largest consumer, has also shown a decrease in P fertilizer consumption since 2013. Although P fertilizer application decreases, crop yield and P uptake may not decline correspondingly. The large P surpluses due to previous P application in European countries have built up soil residual P pool which can continually supply crop production (Sattari et al., 2012). Similarly, the reduced usage

495

of P fertilizer in China may also result from the supply of soil legacy P resources accumulated over the last decades. The increased crop yield and decreased P fertilizer usage indicated an enhancement in P use efficiency in these industrialized countries. Meanwhile, many developing countries, such as India and Brazil, continue to experience elevated P fertilizer usage and an amplified P surplus on croplands (Zhang et al., 2017b). Utilizing residual soil P can be a key strategy to reduce reliance on imported mineral P fertilizer and improve the sustainability of agriculture in these countries.

### 4.3 Limitations in Data Development

Despite numerous improvements in the HaPi dataset over previous P input datasets, several limitations remain in its development. We calculated manure P inputs based on manure N inputs and P:N ratio with spatiotemporal heterogeneity, but the gridded manure P data were not constrained by survey data, as country-level manure P inputs data were not available from the Soil Nutrient Budget database in FAOSTAT. The baseline crop-specific fertilizer rate used to develop the fertilizer input data is assumed to be constant within each country, thereby disregarding regional variability in fertilizer application rates for the same crop type. We used historical cropland, pasture, and rangeland data from HYDE/LUHv2 to spatialize country-level P fertilizer use amounts, but HYDE/LUHv2 data have been shown to exhibit inconsistent spatial and temporal patterns of land use relative to satellite-derived land use at the regional scale. For example, the HYDE dataset overestimates the cropland area in India, which can lead to the underestimation of the P fertilizer application rate on cropland. In this study, we assumed that the ratios of P fertilizer application on pasture relative to cropland were the same as those for N fertilizer, which may not accurately reflect the actual allocation of P fertilizer usage on pasture. Since country-level data for P fertilizer are only available from 1961 onwards, we assumed that the change rates of global fertilizer inputs before 1961 followed the annual global trends reported by Cordell et al. (2009). This assumption ignores regional variations in the changes of P fertilizer usage across

522523

527528

534535

543544

different countries before 1961. Aside from mineral fertilizer, livestock manure, and atmospheric deposition, other anthropogenic P sources, including guano, livestock bones, and human excreta, were historically used to enhance soil fertility. These P sources were difficult to quantify at a global level without reliable data sources and were therefore not included or accurately quantified in the HaPi dataset.

#### 4.4 Uncertainty

The uncertainties in the HaPi dataset mainly arise from the input datasets and methodological assumptions used in developing the global P input estimates. Quantifying the overall uncertainties in the HaPi dataset is challenging due to the heterogeneity of underlying data sources and the scarcity of independent datasets for robust validation. Four major sources of uncertainty were identified as contributing to the overall uncertainty in the HaPi dataset. First, the FAO survey data, which serve as the primary constraint for national total P inputs, represents a critical source of uncertainty in estimating total P inputs to the terrestrial biosphere. Based on expert judgment, a generic uncertainty of approximately 20% was assigned to the FAO national estimates. Consequently, the spatially explicit P input maps derived from these national data inherently carry at least the same level of uncertainty, with a minimum estimated uncertainty of 20%. Second, uncertainties arise from the land use and crop rotation distribution maps used to spatialize P inputs. Uncertainties in land cover classification or temporal interpolation in HYDE/LUHv2 data may propagate into the spatial allocation of P fertilizer and manure applications, thereby affecting local input rates and hotspot identification. Third, uncertainties stem from empirical assumptions and static parameters used during dataset construction, including the assumption of partitioning ratios for fertilizer use between cropland and pasture, the use of timeinvariant crop-specific fertilizer and manure application patterns, and the application of globally uniform change rates for reconstructing pre-1961 data. Finally, uncertainties in atmospheric P deposition simulated by the GEOS-Chem model are primarily associated with model structure and parameterization. Collectively, these sources

548 contribute to uncertainties in both the magnitude and spatial distribution of anthropogenic nitrogen inputs. Given the limitations and uncertainties in the HaPi 549 dataset, it is important to collect or conduct surveys of crop-specific P fertilizer and 550 551 manure use at subnational scales and update global land use data to reflect more precise regional patterns of global fertilizer and manure P inputs. 552 553 554 Data availability The History of Anthropogenic P Inputs (HaPi) dataset is available at 555 556 https://doi.org/10.6084/m9.figshare.29930279.v1 (Bian et al., 2025). 557 558 **Competing interests** 559 At least one of the (co-)authors is a member of the editorial board of Earth System Science Data. The authors have no other competing interests to declare. 560 561 562 **Author contributions** 563 Z.B., H.S., and H.T. designed this work and developed the datasets. R.L., S.Y., B.H., 564 and J.M. estimated atmospheric P deposition. F.N.T. provided the FAO dataset. N.D.M. 565 provided the crop-specific P fertilizer application datasets. L.F., H.R., L.L., H.C., and 566 Y.Y. contributed to the development of methodology. All authors contributed to the 567 writing of the manuscript. 568 569 Acknowledgements 570 Z.B. acknowledges funding support from the National Natural Science Foundation of 571 572 China (42401015; U24A20577) and Jiangsu Provincial Department of Science and Technology (BK20240599). H.S. received support from the Joint CAS-MPG Research 573

- Project (HZXM20225001MI) and the National Key Research and Development
- Program of China (Grant No. 2023YFF1303700). H.T. acknowledges the funding
- support from U.S. National Science Foundation (Grant No. 1903722) and USDA CBG
- (Grant No. TENX12899). FAOSTAT data are produced under FAO regular budget, with
- inputs from national experts in member countries. The views expressed in this work are
- the authors' only and do not reflect FAO positions on the subject matter.

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

© Author(s) 2025. CC BY 4.0 License.

- Feng, X., Lin, H., Fu, T.-M., Sulprizio, M. P., Zhuang, J., Jacob, D. J., Tian, H., Ma, Y., Zhang,
- 643 L., Wang, X., Chen, Q., and Han, Z.: WRF-GC (v2.0): online two-way coupling of WRF
- (v3.9.1.1) and GEOS-Chem (v12.7.2) for modeling regional atmospheric chemistry-
- meteorology interactions, Geosci. Model Dev., 14, 3741–3768, https://doi.org/10.5194/gmd-
- 14-3741-2021, 2021.
- Fick, S. E. and Hijmans, R. J.: WorldClim 2: new 1-km spatial resolution climate surfaces for
- global land areas, Int. J. Climatol., 37, 4302–4315, https://doi.org/10.1002/joc.5086, 2017.
- Fink, G., Alcamo, J., Flörke, M., and Reder, K.: Phosphorus Loadings to the World's Largest
- Lakes: Sources and Trends, Glob. Biogeochem. Cycles, 32, 617-634,
- https://doi.org/10.1002/2017GB005858, 2018.
- Ge, C., Wang, J., Carn, S., Yang, K., Ginoux, P., and Krotkov, N.: Satellite-based global
- volcanic SO2 emissions and sulfate direct radiative forcing during 2005–2012, J. Geophys. Res.
- Atmospheres, 121, 3446–3464, https://doi.org/10.1002/2015JD023134, 2016.
- Gilbert, M., Nicolas, G., Cinardi, G., Van Boeckel, T. P., Vanwambeke, S. O., Wint, G. R. W.,
- and Robinson, T. P.: Global distribution data for cattle, buffaloes, horses, sheep, goats, pigs,
- chickens and ducks in 2010, Sci. Data, 5, 180227, https://doi.org/10.1038/sdata.2018.227, 2018.
- Hurtt, G. C., Chini, L., Sahajpal, R., Frolking, S., Bodirsky, B. L., Calvin, K., Doelman, J. C.,
- Fisk, J., Fujimori, S., Klein Goldewijk, K., Hasegawa, T., Havlik, P., Heinimann, A.,
- Humpenöder, F., Jungclaus, J., Kaplan, J. O., Kennedy, J., Krisztin, T., Lawrence, D., Lawrence,
- P., Ma, L., Mertz, O., Pongratz, J., Popp, A., Poulter, B., Riahi, K., Shevliakova, E., Stehfest,
- E., Thornton, P., Tubiello, F. N., van Vuuren, D. P., and Zhang, X.: Harmonization of global
- land use change and management for the period 850-2100 (LUH2) for CMIP6, Geosci. Model
- Dev., 13, 5425–5464, https://doi.org/10.5194/gmd-13-5425-2020, 2020.
- Klein Goldewijk, K., Beusen, A., Doelman, J., and Stehfest, E.: Anthropogenic land use
- estimates for the Holocene HYDE 3.2, Earth Syst. Sci. Data, 9, 927-953,
- https://doi.org/10.5194/essd-9-927-2017, 2017.
- Lassaletta, L., Billen, G., Grizzetti, B., Anglade, J., and Garnier, J.: 50 year trends in nitrogen
- use efficiency of world cropping systems: the relationship between yield and nitrogen input to
- cropland, Environ. Res. Lett., 9, 105011, https://doi.org/10.1088/1748-9326/9/10/105011, 2014.
- Li, R., Gao, Y., Zhang, L., Shen, Y., Xu, T., Sun, W., and Wang, G.: Global estimates of ambient
- reactive nitrogen components during 2000–2100 based on the multi-stage model, Atmospheric
- Chem. Phys., 24, 7623–7636, https://doi.org/10.5194/acp-24-7623-2024, 2024.
- Lu, C. C. and Tian, H.: Global nitrogen and phosphorus fertilizer use for agriculture production
- in the past half century: shifted hot spots and nutrient imbalance, Earth Syst. Sci. Data, 9, 181,
- 2017.
- Lun, F., Liu, J., Ciais, P., Nesme, T., Chang, J., Wang, R., Goll, D., Sardans, J., Peñuelas, J., and
- Obersteiner, M.: Global and regional phosphorus budgets in agricultural systems and their

- implications for phosphorus-use efficiency, Earth Syst. Sci. Data, 10, 1-18,
- https://doi.org/10.5194/essd-10-1-2018, 2018.
- 681 Ma, J., Shi, H., Zhu, Y., Li, R., Wang, S., Lu, N., Yao, Y., Bian, Z., and Huang, K.: The Evolution
- of Global Surface Ammonia Concentrations during 2001-2019: Magnitudes, Patterns, and
- Drivers, Environ. Sci. Technol., 59, 5066–5079, https://doi.org/10.1021/acs.est.4c14020, 2025.
- Malone, B. P., McBratney, A. B., Minasny, B., and Wheeler, I.: A general method for
- downscaling earth resource information, Comput. Geosci., 41, 119-125,
- https://doi.org/10.1016/j.cageo.2011.08.021, 2012.
- Meng, J., Martin, R. V., Ginoux, P., Hammer, M., Sulprizio, M. P., Ridley, D. A., and van
- Donkelaar, A.: Grid-independent high-resolution dust emissions (v1.0) for chemical transport
- models: application to GEOS-Chem (12.5.0), Geosci. Model Dev., 14, 4249-4260,
- https://doi.org/10.5194/gmd-14-4249-2021, 2021.
- Mueller, N. D., Gerber, J. S., Johnston, M., Ray, D. K., Ramankutty, N., and Foley, J. A.:
- Closing yield gaps through nutrient and water management, Nature, 490, 254-257,
- https://doi.org/10.1038/nature11420, 2012.
- Muñoz-Sabater, J., Dutra, E., Agustí-Panareda, A., Albergel, C., Arduini, G., Balsamo, G.,
- Boussetta, S., Choulga, M., Harrigan, S., Hersbach, H., Martens, B., Miralles, D. G., Piles, M.,
- Rodríguez-Fernández, N. J., Zsoter, E., Buontempo, C., and Thépaut, J.-N.: ERA5-Land: a
- state-of-the-art global reanalysis dataset for land applications, Earth Syst. Sci. Data, 13, 4349–
- 4383, https://doi.org/10.5194/essd-13-4349-2021, 2021.
- Myriokefalitakis, S., Nenes, A., Baker, A. R., Mihalopoulos, N., and Kanakidou, M.:
- Bioavailable atmospheric phosphorous supply to the global ocean: a 3-D global modeling study,
- Biogeosciences, 13, 6519–6543, https://doi.org/10.5194/bg-13-6519-2016, 2016.
- Nguyen, T. H., Tang, F. H. M., Conchedda, G., Casse, L., Obli-Laryea, G., Tubiello, F. N., and
- Maggi, F.: NPKGRIDS: a global georeferenced dataset of N, P2O5, and K2O fertilizer
- application rates for 173 crops, Sci. Data, 11, 1179, https://doi.org/10.1038/s41597-024-04030-
- 4, 2024.
- Randerson, J. T., Van Der Werf, G. R., Giglio, L., Collatz, G. J., and Kasibhatla, P. S.: Global
- Fire Emissions Database, Version 4.1 (GFEDv4), ORNL DAAC,
- https://doi.org/10.3334/ORNLDAAC/1293, 2015.
- Randles, C. A., Silva, A. M. da, Buchard, V., Colarco, P. R., Darmenov, A., Govindaraju, R.,
- Smirnov, A., Holben, B., Ferrare, R., Hair, J., Shinozuka, Y., and Flynn, C. J.: The MERRA-2
- Aerosol Reanalysis, 1980 Onward. Part I: System Description and Data Assimilation
- Evaluation, https://doi.org/10.1175/JCLI-D-16-0609.1, 2017.
- Ringeval, B., Augusto, L., Monod, H., van Apeldoorn, D., Bouwman, L., Yang, X., Achat, D.
- 714 L., Chini, L. P., Van Oost, K., Guenet, B., Wang, R., Decharme, B., Nesme, T., and Pellerin, S.:
- Phosphorus in agricultural soils: drivers of its distribution at the global scale, Glob. Change

- Biol., 23, 3418–3432, https://doi.org/10.1111/gcb.13618, 2017.
- Ringeval, B., Demay, J., Goll, D. S., He, X., Wang, Y.-P., Hou, E., Matej, S., Erb, K.-H., Wang,
- R., Augusto, L., Lun, F., Nesme, T., Borrelli, P., Helfenstein, J., McDowell, R. W., Pletnyakov,
- P., and Pellerin, S.: A global dataset on phosphorus in agricultural soils, Sci. Data, 11, 17,
- https://doi.org/10.1038/s41597-023-02751-6, 2024.
- Roudier, P., Malone, B. P., Hedley, C. B., Minasny, B., and McBratney, A. B.: Comparison of
- regression methods for spatial downscaling of soil organic carbon stocks maps, Comput.
- Electron. Agric., 142, 91–100, https://doi.org/10.1016/j.compag.2017.08.021, 2017.
- Sattari, S. Z., Bouwman, A. F., Giller, K. E., and van Ittersum, M. K.: Residual soil phosphorus
- as the missing piece in the global phosphorus crisis puzzle, Proc. Natl. Acad. Sci., 109, 6348–
- 6353, https://doi.org/10.1073/pnas.1113675109, 2012.
- Sattari, S. Z., Bouwman, A. F., Martinez Rodríguez, R., Beusen, A. H. W., and van Ittersum, M.
- 728 K.: Negative global phosphorus budgets challenge sustainable intensification of grasslands, Nat.
- Commun., 7, 10696, https://doi.org/10.1038/ncomms10696, 2016.
- Stackpoole, S. M., Stets, E. G., and Sprague, L. A.: Variable impacts of contemporary versus
- legacy agricultural phosphorus on US river water quality, Proc. Natl. Acad. Sci., 116, 20562-
- 20567, https://doi.org/10.1073/pnas.1903226116, 2019.
- Steffen, W., Richardson, K., Rockstrom, J., Cornell, S. E., Fetzer, I., Bennett, E. M., Biggs, R.,
- Carpenter, S. R., de Vries, W., de Wit, C. A., Folke, C., Gerten, D., Heinke, J., Mace, G. M.,
- Persson, L. M., Ramanathan, V., Reyers, B., and Sorlin, S.: Planetary boundaries: Guiding
- human development on a changing planet, Science, 347, 1259855-1259855,
- https://doi.org/10.1126/science.1259855, 2015.
- Tian, H., Bian, Z., Shi, H., Qin, X., Pan, N., Lu, C., Pan, S., Tubiello, F. N., Chang, J.,
- Conchedda, G., Liu, J., Mueller, N., Nishina, K., Xu, R., Yang, J., You, L., and Zhang, B.:
- History of anthropogenic Nitrogen inputs (HaNi) to the terrestrial biosphere: a 5 arcmin
- resolution annual dataset from 1860 to 2019, Earth Syst. Sci. Data, 14, 4551-4568,
- https://doi.org/10.5194/essd-14-4551-2022, 2022.
- Tian, H., Pan, N., Thompson, R. L., Canadell, J. G., Suntharalingam, P., Regnier, P., Davidson,
- E. A., Prather, M., Ciais, P., Muntean, M., Pan, S., Winiwarter, W., Zaehle, S., Zhou, F., Jackson,
- R. B., Bange, H. W., Berthet, S., Bian, Z., Bianchi, D., Bouwman, A. F., Buitenhuis, E. T.,
- Dutton, G., Hu, M., Ito, A., Jain, A. K., Jeltsch-Thömmes, A., Joos, F., Kou-Giesbrecht, S.,
- Krummel, P. B., Lan, X., Landolfi, A., Lauerwald, R., Li, Y., Lu, C., Maavara, T., Manizza, M.,
- Millet, D. B., Mühle, J., Patra, P. K., Peters, G. P., Qin, X., Raymond, P., Resplandy, L.,
- Rosentreter, J. A., Shi, H., Sun, Q., Tonina, D., Tubiello, F. N., van der Werf, G. R., Vuichard,
- 750 N., Wang, J., Wells, K. C., Western, L. M., Wilson, C., Yang, J., Yao, Y., You, Y., and Zhu, Q.:
- Global nitrous oxide budget (1980-2020), Earth Syst. Sci. Data, 16, 2543-2604,
- https://doi.org/10.5194/essd-16-2543-2024, 2024.

- Weng, H., Lin, J., Martin, R., Millet, D. B., Jaeglé, L., Ridley, D., Keller, C., Li, C., Du, M.,
- and Meng, J.: Global high-resolution emissions of soil NOx, sea salt aerosols, and biogenic
- volatile organic compounds, Sci. Data, 7, 148, https://doi.org/10.1038/s41597-020-0488-5,
- 2020.
- Xu, R., Pan, S. F., Chen, J., Chen, G. S., Yang, J., Dangal, S. R. S., Shepard, J. P., and Tian, H.
- Q.: Half-century ammonia emissions from agricultural systems in Southern Asia: Magnitude,
- spatiotemporal patterns, and implications for human health, GeoHealth, 2, 40–53, 2018.
- Yu, Q., You, L., Wood-Sichra, U., Ru, Y., Joglekar, A. K. B., Fritz, S., Xiong, W., Lu, M., Wu,
- 761 W., and Yang, P.: A cultivated planet in 2010 Part 2: The global gridded agricultural-
- production maps, Earth Syst. Sci. Data, 12, 3545–3572, https://doi.org/10.5194/essd-12-3545-
- 2020, 2020.
- Yuan, Z., Jiang, S., Sheng, H., Liu, X., Hua, H., Liu, X., and Zhang, Y.: Human Perturbation of
- the Global Phosphorus Cycle: Changes and Consequences, Environ. Sci. Technol., 52, 2438–
- 2450, https://doi.org/10.1021/acs.est.7b03910, 2018.
- Zhang, B., Tian, H., Lu, C., Dangal, S. R. S., Yang, J., and Pan, S.: Global manure nitrogen
- production and application in cropland during 1860-2014: a 5 arcmin gridded global dataset
- for Earth system modeling, Earth Syst. Sci. Data, 9, 667–678, https://doi.org/10.5194/essd-9-
- 667-2017, 2017a.
- Zhang, J., Beusen, A. H. W., Van Apeldoorn, D. F., Mogollón, J. M., Yu, C., and Bouwman, A.
- F.: Spatiotemporal dynamics of soil phosphorus and crop uptake in global cropland during the
- 20th century, Biogeosciences, 14, 2055–2068, https://doi.org/10.5194/bg-14-2055-2017, 2017b.
- Zou, T., Zhang, X., and Davidson, E. A.: Global trends of cropland phosphorus use and
- sustainability challenges, Nature, 611, 81–87, https://doi.org/10.1038/s41586-022-05220-z,
- 2022.