# Peer review of "History of anthropogenic Phosphorus inputs (HaPi) to the terrestrial"

_Earth System Science Data, 2025_

## Referee Comment (RC1)

Peer Review: Major Revision Recommended

The development of the History of Anthropogenic Phosphorus Inputs (HaPi) dataset presented in this manuscript is a significant undertaking with considerable potential value to the global change research community. The goal of creating a long-term, high-resolution global dataset of P inputs is laudable and addresses a genuine need.

However, from the perspective of remote sensing ecology and spatial data science, several fundamental issues regarding the choice of foundational spatial data, the realism of key temporal assumptions, and the strategy for spatial validation must be addressed before the dataset can be considered reliable for regional-scale applications. These concerns form the basis for my recommendation of major revision.

1. Core Issue: Foundational Flaws in Land Use/Land Cover Data

This is the most critical weakness, as it undermines the entire spatial allocation algorithm.

Problem: The manuscript relies exclusively on the model-simulated and historically reconstructed HYDE 3.2 and LUHv2 datasets for land use, without any cross-validation or correction against satellite-observed land cover data.

Technical Rationale:

Known Biases vs. Remote Sensing Observations: HYDE and LUHv2, particularly for historical periods, are known to contain regional inaccuracies. The authors mention this lightly in Section 4.3 as a "limitation," using the overestimation of cropland in India as an example. This is not a minor limitation but a core source of systemic error. In regions like Africa and South America, historical reconstructions of cropland and pasture are highly uncertain. Distributing P inputs based on biased land area data can lead to severe distortions in the spatial patterns and trends of input fluxes.

Missed Opportunity to Leverage Remote Sensing Products: For the post-1980 period, multiple global Land Use/Land Cover Change (LUCC) products derived from satellite observations exist (e.g., ESA CCI-LC, MODIS Land Cover). The manuscript makes no attempt to use these observational data to constrain or improve the spatial allocation for the last 40 years, which reduces the dataset's credibility for the modern era.

Actionable Suggestions:

1.It is strongly recommended to perform a consistency assessment and, if possible, a correction of the HYDE/LUHv2 cropland and pasture distributions for the period after 1980 using at least one remote sensing-based LUCC product (e.g., ESA CCI-LC).

The discussion must be substantially expanded to analyze how potential land use biases might affect the main conclusions, such as the shifts in regional hotspots.

2. Critical Assumption: The Unrealistic Temporal Invariance of Fertilizer Application Rates

Problem: A fixed, crop-specific fertilizer application rate circa 2000 ($F_{crop,i,2000}$) is applied to the entire period from 1860 to 2020.

Technical Rationale:

Contradicts Agricultural Practices: Agricultural technology, economic conditions, and environmental policies have changed dramatically over 160 years. The fertilization levels of the year 2000 are entirely unrepresentative of early Green Revolution or 19th-century practices. This assumption likely severely underestimates the phosphorus use efficiency of early agriculture while overestimating early application intensities, thereby distorting the historical trajectory of P inputs.

Undermines the Dynamic Crop Rotation Innovation: While the paper rightly highlights "dynamic crop rotation" as an advance, using a static application rate significantly diminishes the impact of this "dynamic" element. The primary driver becomes changes in crop area, not the evolution of management intensity.

Actionable Suggestions:

A sensitivity analysis of this critical assumption is mandatory. A simple scenario (e.g., assuming a linear increase in application rates from a baseline in 1860 to the value in 2000) should be designed to quantify its impact on global and regional P input trends.

In the discussion, this point must be elevated from a "limitation" to a "major source of uncertainty," explicitly acknowledging how it may lead to over- or underestimations in early historical inputs.

3. Insufficient Validation: Lack of Spatially-Explicit Validation Against Remote Sensing or Independent Data

Problem: The validation is confined to comparing global totals with previous studies (Table 3), which only verifies that the "magnitude is reasonable." For a high-resolution spatial dataset, the accuracy of its spatial patterns is paramount, and this is entirely unaddressed.

Technical Rationale:

Underutilization of Remote Sensing Proxies: While soil P cannot be directly sensed, several remote sensing proxy variables correlated with P inputs can be used for indirect validation.

Agricultural Intensity: Night-time light data or time series of vegetation indices (e.g., NDVI from MODIS/AVHRR) can identify agricultural hotspots and should be correlated with the spatial patterns of high P input.

Environmental Impact: P inputs drive eutrophication. The estimated P inputs for major watersheds (e.g., Mississippi, Yangtze) should be compared against time series of satellite-derived chlorophyll-a concentration or water turbidity in downstream lakes and coastal waters.

Lack of Sub-national Case Studies: The claim of revealing "subnational heterogeneity" is unsupported. There are no case studies at provincial, state, or watershed scales comparing HaPi against finer-resolution survey data or regional models.

Actionable Suggestions:

A spatial validation analysis must be added. As a minimum, the spatial distribution of HaPi should be overlain and correlated with the remote sensing proxy variables mentioned above for 2-3 representative regions (e.g., US Corn Belt, Western Europe, Eastern China).

It is recommended to seek existing regional or national agricultural P budget studies and aggregate HaPi data to these regions for comparison, assessing its ability to capture inter-regional differences.

4. Overly Qualitative Uncertainty Analysis

Problem: The uncertainty analysis in Section 4.4 identifies sources but concludes with only a generic "minimum of 20%" estimate. This is insufficient for a dataset intended to force numerical models.

Technical Rationale: In remote sensing and geoscience data processing, the goal is to provide spatially explicit, quantitative uncertainty information. For instance, a Monte Carlo approach or error propagation model could be used to generate an uncertainty map accompanying the P input data, synthesizing uncertainties from land use, application rates, etc.

Actionable Suggestions:

The authors are strongly encouraged to attempt a more quantitative uncertainty analysis. If a fully spatialized product is infeasible, a differentiated discussion of uncertainty levels for different periods and regions (e.g., developed vs. developing countries) should be provided.

Ideally, the published dataset should include per-grid-cell uncertainty estimates, which are critical for downstream users employing this data in models.

Conclusion

In summary, while the concept and global totals of the HaPi dataset are impressive, the issues with its spatial data foundation, temporal dynamics representation, and spatial validation significantly constrain its reliability for applications in regional ecology, environmental management, and models coupled with remote sensing. These are not minor textual issues but fundamental methodological concerns affecting the core quality of the dataset.

Therefore, I recommend Major Revision. The authors must undertake substantial revisions to:

Systematically evaluate the land use data against remote sensing products.

1.Quantify the impact of the static fertilizer rate assumption via sensitivity analysis.

2.Provide robust, spatially-explicit validation using remote sensing proxies and regional case studies.

3.Deliver a more quantitative and spatially-informed uncertainty assessment.

4.Only after addressing these core issues can the HaPi dataset become a truly reliable and powerful tool for the remote sensing ecology community and related fields.